# Incidence of Emergency Peripartum Hysterectomy in a Tertiary Obstetrics Unit in Romania

**DOI:** 10.3390/medicina58010111

**Published:** 2022-01-12

**Authors:** Nicolae Gică, Carina Ragea, Radu Botezatu, Gheorghe Peltecu, Corina Gică, Anca Maria Panaitescu

**Affiliations:** 1Department of Obstetrics and Gynecology, Filantropia Clinical Hospital, 11171 Bucharest, Romania; gica.nicolae@umfcd.ro (N.G.); carina.ragea@yahoo.com (C.R.); radu.botezatu@umfcd.ro (R.B.); gheorghe.peltecu@umfcd.ro (G.P.); corina.gica@drd.umfcd.ro (C.G.); 2Department of Obstetrics and Gynecology, Carol Davila University of Medicine and Pharmacy, 11171 Bucharest, Romania

**Keywords:** postpartum hemorrhage, peripartum hysterectomy, life-saving treatment, maternal morbidity, maternal mortality

## Abstract

*Background and Objectives:* Emergency peripartum hysterectomy (EPH) is a life-saving surgical procedure performed when medical and surgical conservative measures fail to control postpartum hemorrhage. The objective of this study was to estimate the incidence of EPH and to determine the factors leading to this procedure and the maternal outcomes. *Materials and Methods:* A retrospective cohort study with all cases of EPH performed at Filantropia Clinical Hospital in Bucharest between January 2012 and May 2021. *Results:* There were 36 EPH, from a total of 36,099 births recorded. The overall incidence of EPH was 0.99 per 1000 deliveries, most cases being related to placenta accreta spectrum disorder and uterine atony. *Conclusions:* Peripartum hysterectomy is associated with an important maternal morbidity rate and severe complications. Efforts should be made to reduce the number of unnecessary cesarean deliveries.

## 1. Introduction

Emergency peripartum hysterectomy (EPH) is performed at the time of delivery or in the immediate postpartum period as a life-saving procedure for situations in which severe obstetrical hemorrhage is refractory to conservative treatment. EPH can follow a cesarean section (CS) or a vaginal delivery. Peripartum HT is associated with significant maternal morbidity and mortality. The reported incidence of EPH varies worldwide from as low as 0.2 per 1000 deliveries in the Northern countries to 10.1 per 1000 deliveries in India [1]. In low and lower-middle-income settings EPH has a higher incidence than in upper-middle and high-income settings [1]. Furthermore, countries with higher rates of CS have a higher incidence of EPH. Rates of EPH in North America are reported to be higher than those in Europe. This may be attributable to the lower CS rate and higher vaginal birth after CS (VBAC) in Europe [2,3]. The association between the rising CS rates and the necessity for EPH is attributable mostly to the occurrence of placental pathologies, such as placenta praevia and placenta accreta, increta, and percreta (placenta accrete spectrum, PAS) [2,4,5]. The incidence of previous CS in patients with PAS has been reported at 59.8%, whereas in patients with placenta praevia the incidence reaches 75% [6].

The practice of obstetrics has changed over the last decades, mostly because of the significant rise in CS but also due to the introduction of new treatments for obstetrical emergencies. It is possible that the rising trend in CS may indirectly increase the incidence of EPH. This may be concerning, especially for countries where CS rates are high because EPH is associated with a high morbidity rate and loss of fertility.

This study aimed to determine the incidence of EPH in a tertiary obstetrics unit and to describe the factors leading to this decision and the maternal outcomes.

## 2. Materials and Methods

This is a retrospective descriptive cohort study of all cases of EPH performed at Filantropia Clinical Hospital in Bucharest, Romania, between January 2012 and May 2021. Filantropia is a large maternity unity that covers the south-eastern part of Romania where around 4500 deliveries are recorded per year. All deliveries are performed in labor-ward under direct obstetrician supervision. Data was collected from the hospital records and the study was approved by the local Ethics Committee. All women who underwent EPH within the first 48 h after delivery in our hospital were included in the study. Demographic and clinical data such as age, parity, gestational age at delivery, history of previous CS, and mode of delivery were recorded.

Indication for HT, type of HT (total or subtotal), type of anesthesia, conservative surgical measures applied before EPH, and complications were identified and included for analysis. All cases of placenta previa and PAS were classified into a single group indicated as “abnormal placentation”.

Statistical analysis was carried out using an SPSS version 22. A descriptive analysis was performed, and data is presented as frequency and percentage for categorical variables and median and interquartile range (IQR) for continuous variables.

## 3. Results

During the 10-year period, there were 36 EPH performed. Throughout this time a total of 36,099 births were recorded, of which 56.6% were vaginal deliveries. The overall incidence of EPH was 0.099% (0.99 per 1000 deliveries). The demographic and clinical data of the women are revealed in Table 1.

The mean age of the patients was 35 years (range: 24–43 years), slightly older than the median maternal age for all deliveries recorded in the hospital [7]. The mean gestational age at the time of delivery was 38 weeks (range: 29–40 weeks). Among women who underwent EPH, 16 (44.4%) had a history of a previous one or more CSs.

Regarding the mode of delivery in the index pregnancy, the prevalence of EPH was 29 (80.6%) following cesarean delivery and 7 (19.4%) following vaginal delivery.

The major indication for EPH was abnormal placentation (placenta previa/PAS) which accounted for 26 cases (72.2%). It is important to specify that most patients associated both placental pathologies (*n* = 18, 50%) (placenta previa with PAS), whereas seven women (19.4%) presented only PAS and one woman had only placenta previa (2.7%). The second most common indication for EPH was uterine atony unresponsive to other treatments (*n* = 6, 16.7%). Other indications were uterine rupture (*n* = 1, 2.8%), uterine myoma (*n* = 1, 2.8%), abruptio placentae (*n* = 1, 2.8%), and left uterine pedicle rupture (*n* = 1, 2.8%) (Table 2) (Figure 1). Uterine atony requiring EPH was more common in women 2 or more para (4 of the 6).

Specific aspects related to EPH are presented in Table 3. Total HT was performed in most of the women undergoing EPH. General anesthesia was performed in 24 cases (66.7%) either from the beginning or as a conversion indicated by the intensive care team. The mean estimated blood loss was 2200 mL (500–5500 mL). Of note, total HT was associated with greater blood loss compared to subtotal HT (mean blood loss 2250 mL vs. 1300 mL; *p* = 0.01).

To control the bleeding and to prevent hysterectomy additional surgical procedures were performed before the decision of EPH such as suture of placental bed, uterine arteries ligation, internal iliac arteries ligation, B-Lynch suture, and Bakri balloon placement. Ten out of 36 patients (27.7%) had intraoperative complications the most common being bladder injury in 7 cases (19.4%), followed by 3 cases of infundibulopelvic ligament hematoma (8.3%) that required unilateral adnexectomy. Two patients (5.5%) needed a relaparotomy due to hemoperitoneum caused by vaginal cuff bleeding. There was no maternal death recorded in this cohort. It is important to note that in all women with a previous CS, EPH was indicated because of abnormal placentation, usually highly suspected antenatally by ultrasound scan [8]. In all patients with intraoperative bladder injury after EPH, there was also abnormal placenta adhesion and invasion. A comparison between the patients that underwent total HT versus those with subtotal HT for EPH is given in Table 4. A total HT was aimed in all cases; however, a subtotal HT was performed in cases where the surgery was technically difficult or the patient was unstable. All cases of subtotal HT were performed for abnormal placentation.

## 4. Discussion

The incidence of EPH in our center was 0.99 per 1000 deliveries during a 10-year period. The incidence rates of EPH reflect the epidemiological characteristics of the population studies and the rates of CS delivery. Romania has, unfortunately, one of the highest rates of CS in Europe [9]. In Filantropia hospital, the rate of CS is relatively lower compared to other centers in the country, but still very high when compared to the 15% that the WHO recommends [10]. Continuous efforts are made to reduce the number of CS in nulliparous women, with no obstetrical indication, but also in those with previous CS by encouraging vaginal delivery after CS. The most common indication for EPH in this study was abnormal placentation, followed by uterine atony unresponsive to medical treatments. Several studies in the past years have shown that abnormal placentation was the main factor leading to uncontrolled hemorrhage and EPH, replacing uterine atony as the most frequent cause for EPH [1,2,4,11,12]. This may be due to the increasing success in treating uterine atony with uterotonic agents, embolization, the B-Lynch suture, or selective devascularization and to the fact that the incidence of abnormal placentation is on the rise related to previous CS [2,4,13]. A study by Huque et al. showed that there is a threefold higher risk of peripartum hysterectomy for abnormal placentation than for uterine atony (AOR 3.2, 95% CI: 2.7–3.8) [11].

It is important to note that in this study all women with a history of previous CS that underwent EPH also presented abnormal placentation. It is well known that a scarred uterus following a CS or a curettage represents a risk factor for abnormal placentation [14,15,16]. One study found that women with previous CS have an 8-fold risk of abnormal placentation in the next pregnancies [5]. Furthermore, the number of previous CS is associated with an increased risk of abnormal placentation. The incidence of abnormal placentation increases by 47-fold, from 0.19% for one previous CS to 9.1% for 4 previous CS [17].

Of women requiring EPH, 80.6% delivered by cesarean section and 19.4% delivered vaginally before the procedure was required. This is consistent with previous studies showing that delivering by CS has a fourfold higher risk of requiring an HT as compared to vaginal delivery (AOR 4.3, 95% CI: 3.6–5.0) [3,11,18,19]. Furthermore, it has been demonstrated that women with a history of one previous CS have more than double the risk of EPH in the next pregnancy while those with two or more previous CS have more than 18 times the risk [20]. This clearly reflects the association between having a previous CS and abnormal placentation in consequent pregnancies. Another possible explanation for the increased proportion of CS in the index delivery before EPH is that pregnancies suspected with abnormal placentation before delivery are selected for CS.

The rate of intraoperative complications was 27.7% and two patients (5.5%) needed a post-hysterectomy re-laparotomy because of persisting intra-abdominal bleeding. The most common intraoperative complication was bladder injury (19.4%). This finding is consistent with other studies suggesting that bladder and ureter injuries occur more often because placental pathology is distorting the lower uterine segment and the pelvic anatomy [1,12,21]. In this study, all patients with bladder injury presented with placenta previa and PAS. Three cases (8.3%) needed unilateral adnexectomy because of uncontrolled bleeding of the infundibulopelvic ligament. A review of six studies of peripartum hysterectomy reported similar intraoperative complications rates: cystostomy (6–29%), ureteral injury (2–7%), oophorectomy (6%), and 4–33% cases needed reoperation [22].

EPH is associated with extensive blood loss and subsequent massive blood transfusions. The mean estimated blood loss in this study was 2200 mL, ranging from 500 to 5500 mL. The same results were reported in one study where the mean blood loss was 2210 mL and the mean transfusion was 4 blood units [23]. However, other studies have reported a median estimated blood loss ranging from 2.5 to 7.8 L [24,25]. The relatively low EBL reported in this study possibly reflects adequate surgical skills and experience of the obstetric team. Cases of PAS were also suspected antenatally by ultrasound examination of the placenta and electively delivered and operated.

There was no maternal death reported in this study. A worldwide meta-analysis reported a mortality rate after EPH ranging from none, in some countries, to 59.1% in one Nigerian study. The rate was higher in low and lower-middle-income settings compared with upper-middle and high-income settings (11.9 vs. 2.5 per 100 hysterectomies) [1].

Regarding the type of HT, we mainly performed a total hysterectomy. Total hysterectomy was associated with greater blood loss compared to subtotal HT and similar findings were reported in one study, which suggested that subtotal hysterectomy should be the procedure of choice, in selected cases, when performing EPH because this procedure is quicker and is associated with a lower risk of visceral injuries [16]. In this study, the rate of intraoperative complications was higher for total hysterectomy compared to subtotal hysterectomy (30% vs. 16.6%). Nevertheless, other studies did not find statistically significant differences in morbidity between the two procedures [12,26,27]. Being more beneficial in terms of operating time and lesser blood loss, subtotal HT should be preferred, especially in the setting of acute hemorrhage. However, a total hysterectomy might be the only option in cases with abnormal placentation. Hence, the final decision to perform subtotal or total HT depends on the particularities of each case.

The incidence of EPH reported worldwide varies widely (Table 5).

It is of importance to note that the reported incidence of EPH reflects not only the characteristics of the population and the incidence of CS rates but also the resources of the healthcare systems. In this, the incidence of EPH is similar to maternal mortality and other indicators of maternal morbidity [28]. Higher-income countries report a lower incidence of EPH despite more deliveries by CS, however lower-to-middle income countries (LMICs), where resources are limited, report higher EPH incidence. Possible reasons for these differences may be a delay in referrals and obstructed neglected labor resulting in a uterine rupture in LMICs [29,30,31].

Our study presented the following limitations. Firstly, it was a retrospective study, being disadvantaged by limited data availability and lack of long-term follow-up of women undergoing EPH. Secondly, because the EPH is performed rarely, the number of patients included in our study was relatively small. Furthermore, our study was hospital-based, reflects the experience of a referral center, and may be generalizable only to our institution or similar settings. A review of current institutional protocols and teaching programs is performed annually to reduce risks and complications related to delivery.

## 5. Conclusions

In our setting, the incidence of EPH was 0.99 cases per 1000 deliveries. Abnormal placentation was the leading cause of EPH. When EPH was performed, it was associated with important maternal morbidity. Given the evidence of the association of placental pathology and CS, especially repeat cesarean section, reducing the number of unnecessary cesarean deliveries will limit the risk of EPH.

## Figures and Tables

**Figure 1 medicina-58-00111-f001:**
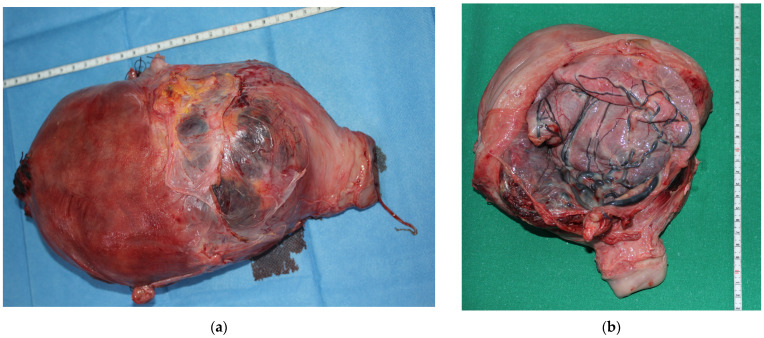
Different aspects of the removed uterus. (**a**) A case of placenta percreta; (**b**) A case of placenta increta; (**c**,**d**) previa uterine myoma.

**Table 1 medicina-58-00111-t001:** Demographic and clinical characteristics of patients that underwent EPH from a total number of 36,099 deliveries.

	Median (Range) or Total Number (%)
**Cases of EPH**	**36**
Incidence of EPH (per 1000 deliveries)	0.099% 0.99 per 1000
**Maternal age at delivery (years)**	**35 (24–43)**
**Parity (median)** ∘ P0 ∘ P1 or more	**2** (1–5) 12 (33.3%) 24 (66.6%)
**Gestational age at delivery (weeks)**	**38 (29–40)**
**History of (caesarean section) CS** ∘ 1 ∘ 2 or more	**16 (44.4%)** 13 (36.1%) 3 (8.3%)
**Mode of delivery in the index pregnancy** ∘ Cesarean delivery ∘ Vaginal delivery	
	29 7

**Table 2 medicina-58-00111-t002:** Indications for emergency peripartum hysterectomy in the 36 patients.

Indication	
**Abnormal placentation** ∘ Placenta accreta spectrum ∘ Placenta praevia ∘ Placenta praevia with placenta accreta spectrum	26 (72.2%) 7 (19.4%) 1 (2.8%) 18 (50%)
**Uterine atony**	6 (16.7%)
**Uterine rupture**	1 (2.8%)
**Uterine myoma**	1 (2.8%)
**Placental abruption**	1 (2.8%)
**Left uterine pedicle rupture**	1 (2.8%)

**Table 3 medicina-58-00111-t003:** Parameters related to peripartum hysterectomy in the 36 cases.

**Type of Hysterectomy** ∘ Total Hysterectomy ∘ Subtotal Hysterectomy	
	30 (83.3%) 6 (16.7%)
**Type of Anesthesia** ∘ General ∘ Spinal	
	24 (66.7%) 12 (33.3%)
**Estimated blood loss (mL)-median (range)**	2200 (500–5500)
**Surgical interventions before hysterectomy**	
∘ Suture of placental bed ∘ Uterine artery ligation ∘ Internal iliac artery ligation ∘ B Lynch suture ∘ Bakri balloon	7 (19.4%) 5 (13.8%) 1 (2.8%) 1 (2.8%) 1 (2.8%)
**Intraoperative complications**	10 (27.7%)
Bladder injury Infundibulopelvic ligament hematoma	7 (19.4%) 3 (8.3%)
**Relaparotomy for vaginal cuff bleeding**	2 (5.5%)
**Maternal mortality**	0

**Table 4 medicina-58-00111-t004:** Comparison of total vs. subtotal emergency peripartum hysterectomy.

Type of Hysterectomy	Total (*n* = 30)	Subtotal (*n* = 6)
Previous CS	11 (36.6%)	5 (83.3%)
Mode of delivery		
∘ Cesarean ∘ Vaginal	23 (76.6%) 7 (23.3%)	6 (100%) 0
Indication of EPH		
Abnormal placentation	21 (70%)	5 (83.3%)
Uterine atony	6 (20%)	0
Uterine rupture	1 (3.3%)	0
Uterine myoma	1 (3.3%)	0
Abruptio placentae	1 (3.3%)	0
Left uterine pedicle rupture	0	1 (16.6%)
Blood loss (mL)-mean (range)	2250 (500–5500)	1300 (500–2200); *p* = 0.01
Intraoperative complications	9 (30%)	1 (16.6%); *p* = 0.03
Bladder injury	6 (20%)	1 (16.6%)
Infundibulopelvic ligament hematoma	3 (10%)	0

**Table 5 medicina-58-00111-t005:** Incidence of EPH around the world per 1000 deliveries.

Country	Incidence of EPH per 1000 Deliveries
Netherland	0.33
Norway	0.20
Ireland	0.30
Israel	0.50
Saudi Arabia	0.63
United States of America	2.7
Romania (this study)	0.99
South-Africa	9.5
Nigeria	4
Pakistan	11
India	10.1

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
