# Peer review of "Incidence of Emergency Peripartum Hysterectomy in a Tertiary Obstetrics Unit in Romania"

_medicina, 2022, doi:10.3390/medicina58010111_

Round 1
Reviewer 1 Report
1. In the methodology: what are the inclusion and exclusion criteria
2. During the 10-year study period, there were 36 EPH performed, with an overall incidence of EPH was 0.099% (0.99 per 1,000 deliveries), as mentioned in line 69. On the last line of the first table, the mode of delivery in the index pregnancy was vaginal delivery in 7 (19.4%). This statistical figure needs a review of the procedure’s total incidence during the study period. The numbers do not match the results.
3. The authors mentioned that a comparison between the patients that underwent total HT versus those with subtotal HT for EPH as reported in table 4. I need clear justification why some patients underwent the subtotal hysterectomy procedure, and what are the values of keeping the cervix in situ?
4. online 135 “A study showed a threefold higher risk of peripartum hysterectomy 135 for abnormal placentation than for uterine atony (AOR 3.2, 95% CI: 2.7–3.8) [11]”. This statement needs to rephrase “A study by…………………..showed that ……………..[11].
5. Since a long time ago, in modern Obstetrics, uterine atony is more ranking as the first cause of postpartum hemorrhage and EPH. It is more reasonable to add This comment in the discussion section, rather than what was written online 135-136.
6. The statement in line 137 “It is important to note that in our study all women with a history of previous cesarean section presented also abnormal placentation”. This is a clear indication to review the protocol management and proper assessment of the teaching skills program.
Author Response
Dear Reviewer,
We would like to thank you very much for taking your time to review our manuscript. Your comments and suggestions are very welcomed and have made a significant improvement to our manuscript.
Please find below the responses to one of each raised points:
- In the methodology: what are the inclusion and exclusion criteria
Thank you for noticing this. We have now added in the Materials and methods section that all cases that required EPH within 48 hours from delivery were included.
- During the 10-year study period, there were 36 EPH performed, with an overall incidence of EPH was 0.099% (0.99 per 1,000 deliveries), as mentioned in line 69. On the last line of the first table, the mode of delivery in the index pregnancy was vaginal delivery in 7 (19.4%). This statistical figure needs a review of the procedure’s total incidence during the study period. The numbers do not match the results.
Thank you for noticing this. While is true that from the 36 cases of EPH, 29 were after a caesarean section and 7 after a vaginal delivery, we acknowledge that the percentages may be confusing for the readers and we therefore removed them.
- The authors mentioned that a comparison between the patients that underwent total HT versus those with subtotal HT for EPH as reported in table 4. I need clear justification why some patients underwent the subtotal hysterectomy procedure, and what are the values of keeping the cervix in situ?
That is a great point you have raised, thank you. We have clarified now Lines 113-114 that in all cases we have aimed for a total hysterectomy but, because of difficulties during surgery and because an unstable patient, subtotal HT was performed in 6 cases. We have clarified now that all these cases were with abnormal placentation (PAS).
- online 135 “A study showed a threefold higher risk of peripartum hysterectomy 135 for abnormal placentation than for uterine atony (AOR 3.2, 95% CI: 2.7–3.8) [11]”. This statement needs to rephrase “A study by…………………..showed that ……………..[11].
Thank you, we have rephrased this now.
5. Since a long time ago, in modern Obstetrics, uterine atony is more ranking as the first cause of postpartum haemorrhage and EPH. It is more reasonable to add This comment in the discussion section, rather than what was written online 135-136.
Thank you, we have made it now clear that postpartum hysterectomy is mainly due to uterine atony but also abnormal placentation.
6. The statement in line 137 “It is important to note that in our study all women with a history of previous caesarean section presented also abnormal placentation”. This is a clear indication to review the protocol management and proper assessment of the teaching skills program.
This is a great observation. What our study aims to show is that a previous caesarean section increases the risk of EPH in a subsequent pregnancy. It is true that in our country we have unfortunately a high incidence of caesarean sections. This study underlines the importance of avoiding CS without an obstetrical indication. We have now introduced a discussion point that protocols and teaching skills programs need to be reviewed to improve outcomes and avoid complications. (lines 197-198)
Reviewer 2 Report
Incidence of emergency peripartum hysterectomy in a tertiary obstetrics unit in Romania
Reviewer Comment
This is a great paper, well written and is so important in the Reproductive Health domain to gauge Obstetricians realization that the rising tide of Caesarian section has consequences of abnormal placentation and also other obstetric emergencies such as Uterine atony that contribute to this major obstetric emergency surgical procedure in EPH. Congratulations to the study team and lead author of this paper.
I have a few comments to make!
General Comment
- The study was conducted in a tertiary referral hospital with very low incidence of EPH. The indications of EPH are mostly due to abnormal placentation/PAS and uterine atony. A lot of reference and comparison is being made to higher income countries whose incidences are almost the same or closer. Only inferences from a developing country is with India. It would be nicer to include a paragraph and cite one or two more references for LMIC or resources limited settings to portray the challenges of higher incidences, most commonly due to 3 Delays in referrals and obstructed neglected labour resulting in uterine rupture. This will add flavor in the background and bring out the novelty/ of the current study based in a bigger hospital in a developed world.
- Also, minimize self-citations and report in the third person.
Title
Acceptable
Abstract
Maybe needs to be shortened abit as the overall article is not very long
Line 15. Avoid self-citations.
Introduction
Well presented with good paragraph connections, flow and further strengthened with appropriate references. Ends with aims of the current study introduced well.
Lines 36-39 needs to be rephrased to make it sound clearer.
Methods
The study site background needs to be elaborated a bit more to give readers some perspective .eg, labour ward or delivery suite, type of specialists and annual delivery rate etc.
Results
Results satisfactorily presented. Table title, column heads and data categorised well for clarity. Text results are appropriate and reflective of significance in most cases.
Line 93-95: How much is relatively low blood loss when the average is 2200ml almost half of the total blood volume? This sentence may need to be rephrased.
Elaborating also to certain extent on uterine atony (6 cases) as the second commonest cause for EPH would be worthwhile. Was it more common for grandmulitparity or twins etc. and did not respond to uterotonics?
Discussions
Page 5 line 168 -169 The blood loss compared here is still higher, but in comparison to other settings, this was due to good pre-operative preparation in anticipation for problem as cases were diagnosed earlier with scans and I assume that most of the CS were done as elective. Also as well as possibly personnel expertise.
Line 69: remove ‘our’ avoid self-citations
Conclusions
Well written
References:
Reference syntax and spelling are consistent and adequate number.
Author Response
This is a great paper, well written and is so important in the Reproductive Health domain to gauge Obstetricians realization that the rising tide of Caesarean section has consequences of abnormal placentation and also other obstetric emergencies such as Uterine atony that contribute to this major obstetric emergency surgical procedure in EPH. Congratulations to the study team and lead author of this paper.
Dear Reviewer, we honestly thank you very much for taking time to review our manuscript and for the very important points raised.
Please find below a point-by-point response to the raised issues.
I have a few comments to make!
General Comment
- The study was conducted in a tertiary referral hospital with very low incidence of EPH. The indications of EPH are mostly due to abnormal placentation/PAS and uterine atony. A lot of reference and comparison is being made to higher income countries whose incidences are almost the same or closer. Only inferences from a developing country is with India. It would be nicer to include a paragraph and cite one or two more references for LMIC or resources limited settings to portray the challenges of higher incidences, most commonly due to 3 Delays in referrals and obstructed neglected labour resulting in uterine rupture. This will add flavor in the background and bring out the novelty/ of the current study based in a bigger hospital in a developed world.
Thank you very much, we have now added a paragraph at the end of Discussion section (lines 187-200) where we have addressed the issue of disparities in the global reported incidence of EPH. We moved Table 5 here as we felt it relates more to the discussion in this paragraph.
- Also, minimize self-citations and report in the third person.
Thank you, we have done this throughout the document.
Title
Acceptable
Thank you.
Abstract
Maybe needs to be shortened abit as the overall article is not very long
Line 15. Avoid self-citations.
Thank you, we have shortened the abstract and removed self-citation from line 15.
Introduction
Well presented with good paragraph connections, flow and further strengthened with appropriate references. Ends with aims of the current study introduced well.
Lines 36-39 needs to be rephrased to make it sound clearer.
Thank you, we have rephrased this now.
Methods
The study site background needs to be elaborated a bit more to give readers some perspective .eg, labour ward or delivery suite, type of specialists and annual delivery rate etc.
Thank you, we have added that in the Methods section.
Results
Results satisfactorily presented. Table title, column heads and data categorised well for clarity. Text results are appropriate and reflective of significance in most cases.
Thank you.
Line 93-95: How much is relatively low blood loss when the average is 2200ml almost half of the total blood volume? This sentence may need to be rephrased.
Thank you, we have removed “relatively low”.
Elaborating also to certain extent on uterine atony (6 cases) as the second commonest cause for EPH would be worthwhile. Was it more common for grandmulitparity or twins etc. and did not respond to uterotonics?
Thank you, we have introduced uterine atony as the second cause in the Abstract and detailed on the cases we had in the Results section.
Discussions
Page 5 line 168 -169 The blood loss compared here is still higher, but in comparison to other settings, this was due to good pre-operative preparation in anticipation for problem as cases were diagnosed earlier with scans and I assume that most of the CS were done as elective. Also as well as possibly personnel expertise.
Thank you.
Line 69: remove ‘our’ avoid self-citations
Thank you, we did that throughout the document.
Conclusions
Well written
Thank you.
References:
Reference syntax and spelling are consistent and adequate number.
